# Design, Synthesis, In Vitro Antifungal Activity and Mechanism Study of the Novel 4-Substituted Mandelic Acid Derivatives

**DOI:** 10.3390/ijms24108898

**Published:** 2023-05-17

**Authors:** Biao Chen, Dandan Song, Huabin Shi, Kuai Chen, Zhibing Wu, Huifang Chai

**Affiliations:** 1School of Pharmacy, Guizhou University of Traditional Chinese Medicine, Guiyang 550025, China; gzdxchenbiao@163.com; 2State Key Laboratory Breeding Base of Green Pesticide and Agricultural Bioengineering, Key Laboratory of Green Pesticide and Agricultural Bioengineering, Ministry of Education, Center for R&D of Fine Chemicals of Guizhou University, Guiyang 550025, China; 15284637047@163.com (D.S.); shb1126@163.com (H.S.); m15185174268@163.com (K.C.)

**Keywords:** 4-substituted mandelic acid derivatives, 1,3,4-oxadiazole thioether, antifungal activity, morphological study, cell membrane integrity

## Abstract

Plant diseases caused by phytopathogenic fungi are a serious threat in the process of crop production and cause large economic losses to global agriculture. To obtain high-antifungal-activity compounds with novel action mechanisms, a series of 4-substituted mandelic acid derivatives containing a 1,3,4-oxadiazole moiety were designed and synthesized. In vitro bioassay results revealed that some compounds exhibited excellent activity against the tested fungi. Among them, the EC_50_ values of **E_13_** against *Gibberella saubinetii* (*G. saubinetii*), **E_6_** against *Verticillium dahlia* (*V. dahlia*), and **E_18_** against *Sclerotinia sclerotiorum* (*S. sclerotiorum*) were 20.4, 12.7, and 8.0 mg/L, respectively, which were highly superior to that of the commercialized fungicide mandipropamid. The morphological studies of *G. saubinetii* with a fluorescence microscope (FM) and scanning electron microscope (SEM) indicated that **E_13_** broke the surface of the hyphae and destroyed cell membrane integrity with increased concentration, thereby inhibiting fungal reproduction. Further cytoplasmic content leakage determination results showed a dramatic increase of the nucleic acid and protein concentrations in mycelia with **E_13_** treatment, which also indicated that the title compound **E_13_** could destroy cell membrane integrity and affect the growth of fungi. These results provide important information for further study of the mechanism of action of mandelic acid derivatives and their structural derivatization.

## 1. Introduction

Plant diseases caused by phytopathogenic fungi are a serious threat to the process of crop production and cause large economic losses to global agriculture [1,2]. More than 8000 species of fungi are known to cause plant diseases, such as *Fusarium graminearum* (*F. graminearum*), *Gibberella saubinetii* (*G. saubinetii*), *Sclerotinia sclerotiorum* (*S. sclerotiorum*), *Botrytis cinerea* (*B. cinerea*), etc., and these fungi cause 85% of plant diseases and have caused huge economic crop losses [3]. Moreover, some fungi can also produce toxins that are harmful to animal and human health, such as aflatoxins, fumonisins, and trichocenes, which will last for many years [4,5]. Therefore, the prevention and control of fungal diseases have become an urgent problem in agricultural production. Currently, chemical control is still an effective method to prevent plant fungal diseases. However, the frequent use and abuse of chemical pesticides not only bring about environmental pollution and pesticide residue problems but also accelerate the emergence of resistance in pathogenic fungi [6,7]. At present, it is urgent to develop new antifungal pesticides with high efficiency, novel action mechanism, no cross-resistance, and environmental friendliness [8].

Mandelic acid (MA) has very specific physiological activity and is used as an important intermediate to synthesize medicines, such as cephalosporin antibiotics, vasodilators such as ring mandelic acid, nonsteroidal anti-inflammatory drugs such as norrecoxib and celecoxib, and urinary tract disinfectants such as urotropine [9,10,11]. In terms of pesticides, mandipropamid was the first commercialized mandelic acid fungicide, marketed in 2001, and had a good control effect on *Pseudoperonospora cubensis* and *Pphytophora nicotianae* [12,13]. As reported in recent years, 1,3,4-oxadiazole derivatives exhibit a wide range of excellent pesticide activities, including antibacterial [14,15,16], antifungal [17], anti-TMV [18,19,20], and nematicidal activities (Figure 1) [21], and therefore is widely considered in pesticide design.

To obtain high antifungal activity compounds with a novel action mechanism, a series of 4-substituted mandelic acid derivatives containing a “1,3,4-oxadiazole thioether” were designed and synthesized (Figure 2). In vitro antifngal bioassays indicated that some compounds exhibited excellent antifungal activities against three tested fungi (*G. saubinetii*, *V. dahlia*, and *S. sclerotiorum*). In order to investigate the antifungal mechanism of the highly active compounds, we conducted morphological studies using the compound E_13_. The morphological study of *G. saubinetii* hyphae with the treatment of title compound **E_13_** by FM and SEM revealed that it could remarkably break the surface morphology of mycelia. Furthermore, the results of intracellular nucleic acid and protein leakage experiments further demonstrated that title compound **E_13_** could destroy the integrity of the cell membrane of *G. saubinetii*, thus inhibiting the growth of mycelium. These results provide important clues for the further mechanism study and derivatization of mandelic acid derivatives used for plant disease control.

## 2. Results and Discussion

### 2.1. Chemistry

The synthetic route is depicted in Figure 3. As initial materials, 4-substituted mandelic acids were used to synthesize key intermediate ester **B**. Then, **C** was obtained from **B** through hydrazinolysis in the presence of 80% hydrazine hydrate and reacted with CS_2_ under base conditions to form intermediate thiol **D**. Title compounds **E_1_**–**E_28_** were synthesized from **D** with different substituted benzyl halides with K_2_CO_3_ in cetonitrile and water. The title compounds were characterized by ^1^H nuclear magnetic resonance (NMR), ^13^C NMR, ^19^F NMR spectroscopy, and high-resolution mass spectrometry (HRMS).

### 2.2. Crystal Structure

The crystal structure of title compound **E_9_** was confirmed by X-ray single-crystal diffraction analysis (Figure 4). A combination of absorption correction and Lorentz polarization was adopted. Structural solutions and F_2_-based full-matrix least-squares refinements were conducted using SHELXT-14 software. An isotropic refinement of the nonhydrogen atoms was then carried out. Subsequently, the scattering factors of the neutral atoms were analyzed by incorporating an anomalous dispersion correction. The majority of H_2_O molecules in the crystalline structure were removed using the SQUEEZE option in PLATON.

### 2.3. In Vitro Antifungal Activity Analysis

The preliminary antifungal activity results of the title compounds against seven plant phytogenic fungi at 100 mg/L are shown in Table 1. As indicated, some title compounds exhibited excellent inhibitory activities against the seven tested fungi at a concentration of 100 mg/L. In detail, compounds E_10_, E_13,_ and E_14_ showed good antifungal activity against *G. saubinetii*, with inhibition rates of 78.5%, 78.3%, and 80.2%, respectively, which were higher than mandipropamid (19.1%). The inhibition rates of E_6_, E_10_, E_13_, E_14,_ and E_20_ against *V. dahliae* were 93.8%, 91.8%, 100%, 95.9%, and 94.4%, respectively, which were much higher than mandipropamid (27.2%). Compounds E_9_, E_13,_ and E_17_ exhibited 92.8%, 92.5%, and 89.7% inhibition of *S. sclerotiorum*, respectively, which was much better than that of the commercialized fungicide mandipropamid (14.0%). Compounds E_10_, E_13,_ and E_14_ showed good antifungal activity against *Fusarium oxysporum*, with inhibition rates of 69.4%, 70.3%, and 68.8%, respectively, which were higher than mandipropamid (26.5%). Compounds E_5_, E_7_, E_10,_ and E_13_ showed remarkable antifungal activity against *T. cucumeris*, with inhibition rates of 84.9%, 81.4%, 81.1%, and 87.5%, respectively, which were higher than hymexazol (79.3%). These results indicate the broad-spectrum inhibitory activities of 4-substituted mandelic acid derivatives. As shown in Table 2, some title compounds showed obviously superior antifungal activity than commercial pesticides. For example, the EC_50_ values of E_13_, E_14,_ and E_17_ against *G. saubinetii* were 20.4, 21.5, and 22.0 mg/L, respectively; the EC_50_ values of E_6_, E_13,_ and E_18_ against *V. dahlia* were 12.7, 18.5, and 15.7 mg/L, respectively; the EC_50_ values of E_8_, E_9_, and E_18_ against *S. sclerotiorum* were 13.8, 10.3, and 8.0 mg/L, respectively; the EC_50_ values of E_7_ and E_13_ against *T. cucumeris* were 5.7 and 7.1 mg/L, respectively; which were much higher than corresponding control fungicides. The preliminary structure-activity relationship analysis showed that when R_2_ were the same substituted phenyl groups and R_1_ were electron-withdrawing groups, the anti-*G. saubinetii* and anti-*S. sclerotiorum* activities were higher than electron-donating groups, such as E_9_ > E_2_, E_13_/E_20_ > E_6_, E_14_/E_21_ > E_7_. When R_1_ was the same group and R_2_ was substituted phenyl groups, the anti-*G. saubinetii* and anti-*S. sclerotiorum* activities were higher than that of non-substituted phenyl groups, such as E_2_/E_7_ > E_1_, E_9_/E_10_/E_13_/E_14_ > E_8_. In addition, when R_1_ is a halogen group and the para-substituent group or ortho-substituted group of R_2_ are halogen groups, the anti-*G. saubinetii* and anti-*S. sclerotiorum* activities were higher than electron-donating groups, such as E_13_/E_14_ > E_9_, E_24_/E_25_ > E_22_.

### 2.4. Morphological Study of G. saubinetii Fungus with SEM

To investigate the antifungal mechanism, we conducted morphological studies using the compound E_13_. The morphology of *G. saubineti* treated with title compound **E_13_** was observed by SEM (Figure 5). As shown in Figure 5A, hyphae grew normally with the treatment of 1% DMSO (CK group), and the morphology was relatively complete. When treated with the compound **E_13_** at the concentration of 2 eq. EC_50_, the surface of the mycelia began to shrink obviously, as shown in the red circle in Figure 5B. With the concentration of **E_13_** increased to 4 eq. EC_50_, the change in the mycelia morphology intensified, and some holes appeared on the surface of the mycelia, which were marked by red arrows in Figure 5C. When the concentration reached 8 eq. EC_50_, all the mycelia were contracted, and some surfaces were seriously damaged, as shown in the red circle in Figure 5D. We speculated that compound **E_13_** could destroy the cell membrane and wall of *G. saubineti*, culminating in the death of hyphae.

### 2.5. Morphological Study of G. saubinetii Fungus with FM

To further determine whether the membrane of mycelium was damaged, propidium iodide (PI) dye was used to stain the DNA, and fluorescence microscopy (FM) was used to observe its fluorescence. PI can enter the cell across the damaged cellular membrane and specifically bind with DNA to emit red fluorescence [22]. As shown in panels A, B, and C of Figure 6, the cellular membrane of the hyphal cells without staining was colorless. However, when compared with the blank control (Figure 6D), strong fluorescence intensity was observed by FM in the *G. saubinetii* hyphae treated with E_13_ at a concentration of 2 eq. and 4 eq. EC_50_ (Figure 6E,F). Moreover, as shown in Figure 6E,F, the higher the concentration of E_13_, the more red-colored hyphae appear. Therefore, the FM observation further proved that target compound **E_13_** could destroy the membrane integrity of *G. saubinetii*, and the effect was dose-dependent.

### 2.6. Mechanistic Study of the Antifungal Activity of ***E_13_***

Further cytoplasmic content leakage assay confirmed the membrane of mycelia was damaged. Nucleic acid and protein concentrations in mycelial suspensions can be evaluated by measuring the changes in absorbance at 260 and 280 nm [23]. As shown in Figure 7, compared with the blank control (CK) group, the absorbance values at 260 and 280 nm were significantly increased within 2 h after treatment with 2 eq. and 4 eq. EC_50_ of compound **E_13_** and stabilized at approximately 8 h. Therefore, it can be inferred that the nucleic acids and proteins of *G. saubinetii* mycelia were significantly released after treatment with **E_13_**, and the effect was dose-dependent. This result was consistent with the morphological observation by SEM and FM experiments. When treated with the title compound **E_13_**_,_ the cell membrane integrity was damaged, and the nucleic acids and proteins were released, thereby inhibiting the growth of fungi.

## 3. Materials and Methods

### 3.1. Instruments and Chemicals

^1^H nuclear magnetic resonance (NMR), ^13^C NMR, and ^19^F NMR spectra were acquired on a Bruker 400 NMR spectrometer (Bruker Corporation, Karlsruhe, Germany) with tetramethylsilane (TMS) as the internal standard and DMSO-*d_6_* as the solvent. High-resolution mass spectrometry (HRMS) data were obtained on a Thermo Scientific Q Exactive (Thermo Scientific, Waltham, MA, USA). X-ray crystallographic data were collected by a D8 QUEST (Bruker Corporation, Karlsruhe, Germany). The morphology of the fungus was observed by a Nova Nano SEM450 scanning electron microscope (SEM) instrument (Thermo Fisher Scientific, Waltham, MA, USA) and an Olympus-BX53F fluorescence microscope (FM) (Olympus Ltd., Tokyo, Japan). The cytoplasmic content leakage assay was performed by Cytation™5 multimode readers (BioTek Instruments, Inc., Winooski, VT, USA). HPLC analysis was performed on LC–2030C (Shimadzu, Kyoto, Japan). Melting points were measured with SGW X–4B binocular microscope melting point apparatus (Inesa, Shanghai, China). All reagents and solvents are commercially available with chemical or analytical purity. Benzyl halides were purchased from Shanghai Haohong Scientific Co., Ltd. (Shanghai, China).

### 3.2. Fungi

Gibberella saubinetii (Durieu and Mont.) Sacc., Verticillium dahliae Kleb., Sclerotinia sclerotiorum (Lib.) de Bary, Fusarium oxysporum f. sp. Cucumerinum, Phytophthora capsici Leonian and Thanatephorus cucumeris (A.B. Frank) Donk were purchased from Beijing Beina Chuanglian Biotechnology Institute, China. Fusarium proliferatum (Matsush.) Nirenberg ex Gerlach and Nirenberg was provided by our laboratory (State Key Laboratory Breeding Base of Green Pesticide and Agricultural Bioengineering, Guiyang, China). All these fungi were grown on potato dextrose agar (PDA) plates at 25 ± 1 °C and maintained at 4 °C.

### 3.3. Crystallographic Analysis

A single crystal of title compound **E_9_** suitable for X-ray diffraction analysis was obtained by slow evaporation of DMF solution at room temperature. Crystallographic data of compound **E_9_**: colorless crystal, C_17_H_15_FN_2_O_2_S, M_r_ = 330.37, orthorhombic, space group Pbcn, a = 11.2260 (5) Å, b = 17.9952 (8) Å, c = 7.8108 (3) Å; α = 90°, β = 90°, γ = 90°; μ = 2.018 mm^−1^, V = 1577.89 (12) Å^3^, Z = 4, Dc = 1.391 g cm^−1^, F (000) = 688.0, Reflection collected/Independent reflection measured = 2728/2645, Goodness-of-fit on F^2^ = 1.067, Fine, R_1_ = 0.0435, wR_2_ = 0.1112. The crystallographic data of **E_9_** are provided in Appendix A. The supplementary data for title compound **E_9_** have been deposited in the Cambridge Crystallographic Data Centre (accessed on 18 September 2020, http://www.ccdc.cam.ac.uk/conts/retrieving.html) under deposition number 2050255.

### 3.4. General Procedure for the Synthesis of Intermediate ***B***

Intermediate B was synthesized, as reported previously [24]. A mixture of 4-substituted mandelic acid (0.1 mol), 98% sulfuric acid (0.05 mol), and methanol (50 mL) was stirred at 80 °C for 3 h, and the reaction was monitored by TLC. Then, the solvent was concentrated under reduced pressure to obtain **B**, which was used in the next reaction without further purification.

### 3.5. General Synthesis Procedure for Intermediates ***C*** and ***D***

Intermediate **B** (0.1 mol) and 80% hydrazine hydrate (0.2 mol) were dissolved in methanol (20 mL) and reacted at 100 °C for 30 min. Then, the solvent was concentrated in vacuo to obtain **C**. To a solution of KOH (0.15 mol) in 20 mL ethanol, **C** (0.1 mol) was added, and CS_2_ (0.2 mol) was slowly added dropwise into the solution, reacted at r.t. for 12 h and then reacted at 80 °C for 10 h. The solution was concentrated in vacuo. The pH value was adjusted to 3 to 4 with 2 M hydrochloric acid solution, and the mixture was extracted with ethyl acetate (50 mL × 3) and dried with anhydrous sodium sulfate. The ethyl acetate was removed under reduced pressure to obtain **D**, which was used in subsequent reactions without further purification.

### 3.6. General Synthesis Procedure for Target Compounds ***E_1_***–***E_28_***

To a solution of **D** (4.0 mmol) in CH_3_CN/H_2_O (5.0 mL/5.0 mL), K_2_CO_3_ (8.0 mmol) and different benzyl halides (4.0 mmol) were added and reacted at r.t. for 24 h and then concentrated in vacuo. The residue was purified by column chromatography on silica gel (petroleum ether/ethyl acetate = 20/1~10/1) to obtain the title compounds **E_1_**–**E_28_**, and the physical and spectral data are provided in the Appendix A.

### 3.7. In Vitro Antifungal Activity

The fungicidal activities of **E_1_**–**E_28_** were tested in vitro against seven plant pathogenic fungi (*G. saubinetii*, *V. dahlia*, *S. sclerotiorum*, *F. oxysporum*, *F. proliferatum*, *T. cucumeris*, and *P. capsici*) using a mycelial growth inhibition method [25,26]. The preliminary activity screening concentration of the title compounds was 100 mg/L. The mycelial dishes of fungi that were used for testing were cut from the PDA medium cultivated at 25 ± 1 °C and were approximately 4 mm in diameter; the disks were inoculated in the middle of a PDA plate with a germ-free inoculation needle and then incubated for 3 to 5 days at the same temperature. DMSO (1%) in sterile distilled water served as a blank control (CK), whereas the commercialized fungicides mandipropamid and hymexazol served as the positive controls. Each treatment was conducted in triplicate. When the mycelia of the blank control grew to 6 cm, the diameters of the mycelia treated with the title compounds were recorded. Inhibitory effects on these fungi were calculated using the formula I (%) = [(C − T)/(C − 0.4)] × 100, where C represents the diameter of fungal growth of the blank control, T represents the diameter of the fungi with the treated compound, and I represents the inhibition rate. Standard deviation (SD) values were calculated based on the inhibition data of three repetitions for each test compound. Based on the in vitro antifungal activity results, the median effective concentrations (EC_50_) values of the highly active compounds were further determined according to the method described above. A series of activity screening concentrations of the title compounds and positive control consisting of 100, 50, 25, 12.5 or 6.25 mg/L were prepared. EC_50_ values were calculated by linear regression analysis with SPSS software 20.0. The regression equations of the title compounds are provided in Appendix A.

### 3.8. Morphological Observation of G. saubinetii by Scanning Electron Microscopy

To observe the effects of the high-antifungal-activity title compounds on the morphology of fungi, the morphology of the hyphae after treatment with the compound **E_13_** was observed by SEM on the basis of previous methods [27,28,29]. For the SEM experiments, six fungal cakes (4.0 mm) of the *G. saubinetii* were taken and incubated in PDB (potato dextrose broth) medium at 28 °C and 180 rpm for 24 h, and then compound **E_13_** was added to each group at different concentrations (0, 2 eq., 4 eq., and 8 eq. EC_50_), and incubated under the same conditions for 24 h. The mycelia were centrifuged (6000 rpm, 5 min, 4 °C) and washed with PBS buffer (pH 7.2) three times. Then, 2.5% glutaraldehyde was added, and the samples were placed in a refrigerator at 4 °C overnight and dehydrated with gradient ethanol (30%, 50%, 70%, 90%, and 100%) and t-butanol for 15 min. The dehydrated mycelia were dried in a vacuum and sprayed with gold. The mycelia were observed and photographed on a Nova Nano SEM scanning electron microscopy.

### 3.9. Morphological Observation of G. saubinetii by Fluorescence Microscope

According to previously reported methods [30,31], *G. saubinetii* hyphae were cultured in potato dextrose broth (PDB) medium at 28 °C and 180 rpm for 24 h and then treated with 0, 2, and 4 eq. EC_50_ of compound **E_13_** for 24 h. The PDB medium was removed by centrifuging at 4 °C and 6000 rpm for 5 min, and the hyphae were stained with 10 μL of propidium iodide (PI) solution (20 mg/L). The hyphae were incubated at 37 °C for 15 min in the dark and then washed with PBS three times. A coverslip was placed on the hyphae, and the samples were observed and photographed using an Olympus-BX53F fluorescence microscopy.

### 3.10. Determination of Cytoplasmic Content Leakage

The leakage of cytoplasmic contents from *G. saubinetii* mycelia treated with **E_13_** was measured according to the methods described previously with minor modifications [32,33]. Six cakes (4.0 mm) of *G. saubinetii* fungus were placed in 100 mL of PDB at 28 °C for 3 days at 180 rpm. The hyphae were harvested and washed three times with sterile distilled water. The mycelia (4.0 g) were suspended in 20 mL of distilled water containing compound **E_13_** at various concentrations (0, 2 eq., and 4 eq. EC_50_) and incubated on a rotary shaker at 28 °C for 8 h. The absorbance values of the supernatant at 260 nm and 280 nm were determined at 0 h, 2 h, 4 h, 6 h, and 8 h, respectively. Each experiment was conducted in triplicate.

### 3.11. Statistical Analysis

Each experiment was performed in triplicate. The results are expressed as the means ± SD. The EC_50_ values were evaluated with the regression equation and r value. All the data in the same group were evaluated by the Q-test. Differences between groups were compared by one-way analysis of variance (ANOVA; Duncan’s multiple range test) with *p* < 0.05. Statistical tests were performed using the SPSS software package (version 20.0, IBM).

## 4. Conclusions

A series of 4-substituted mandelic acid derivatives containing a 1,3,4-oxadiazole moiety were designed and synthesized. In vitro bioassay results revealed that some title compounds exhibited excellent activity against seven kinds of pathogenic fungi. Among them, the EC_50_ values of **E_13_** against *G. saubinetii*, **E_6_** against *V. dahlia*, and **E_18_** against *S. sclerotiorum* were 20.4, 12.7, and 8.0 mg/L, respectively, which were highly superior to that of the commercialized fungicide mandipropamid. A morphological study of *G. saubinetii* with FM and SEM indicated that **E_13_** broke the surface of the hyphae and destroyed cell membrane integrity with the increased concentration, thereby inhibiting fungal reproduction. Further cytoplasmic content leakage determination results showed a dramatic increase of the nucleic acid and protein concentrations in mycelia with **E_13_** treatment, which also indicated that the title compound **E_13_** could destroy the cell membrane integrity and affect the growth of fungi. Further mechanistic research is planned for future completion.

## Figures and Tables

**Figure 1 ijms-24-08898-f001:**
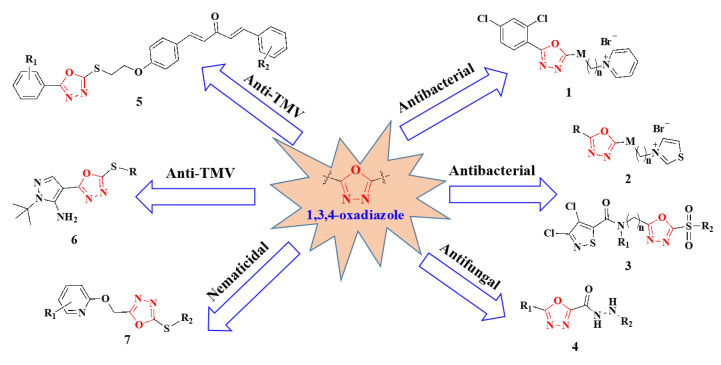
Bioactive structures containing a “1,3,4-oxadiazole” moiety. Compound **1** is from Wang et al., 2016 [14]. Compound **2** is from Wang et al., 2019 [15]. Compound **3** is from Xiang et al., 2020 [16]. Compound **4** is from Wu et al., 2019 [17]. Compound **5** is from Gan et al., 2015 [18] and Gan et al. [19], 2017. Compound **6** is from Yang et al., 2020 [20]. Compound **7** is from Chen et al., 2020 [21].

**Figure 2 ijms-24-08898-f002:**
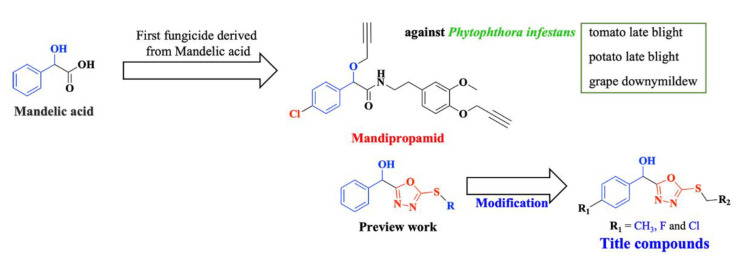
Design and optimization of the title compounds.

**Figure 3 ijms-24-08898-f003:**
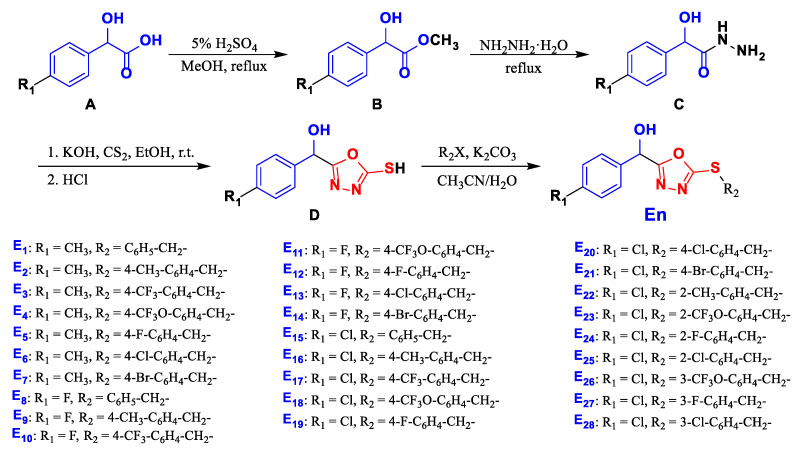
Synthetic route of the title compounds.

**Figure 4 ijms-24-08898-f004:**
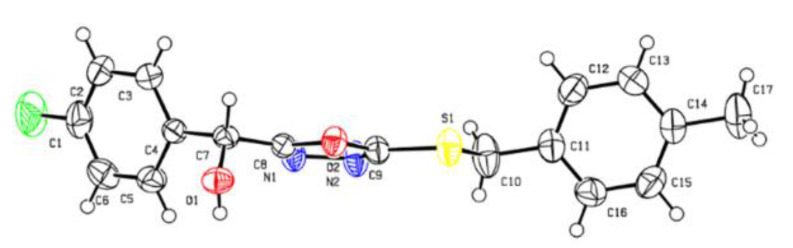
Crystal structure of title compound **E_9_**.

**Figure 5 ijms-24-08898-f005:**
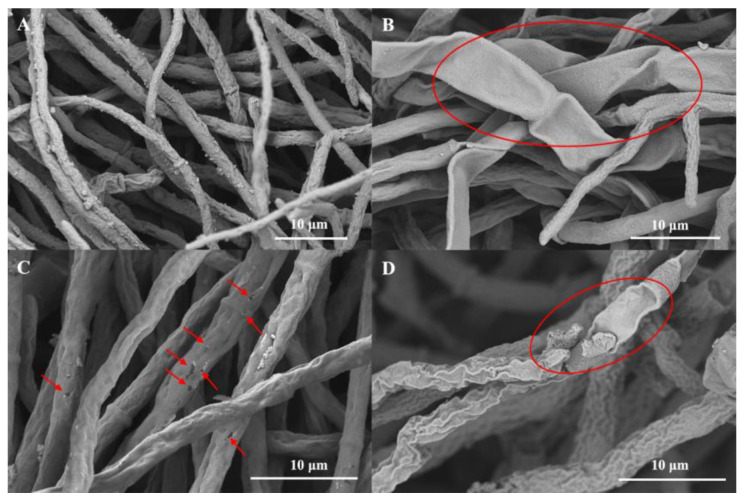
Morphological study of *G. saubinetii* treated with **E_13_** at different dosages in PDB by SEM. (**A**) 0 mg/L (CK); (**B**) 2 eq. EC_50_; (**C**) 4 eq. EC_50_; (**D**) 8 eq. EC_50_.

**Figure 6 ijms-24-08898-f006:**
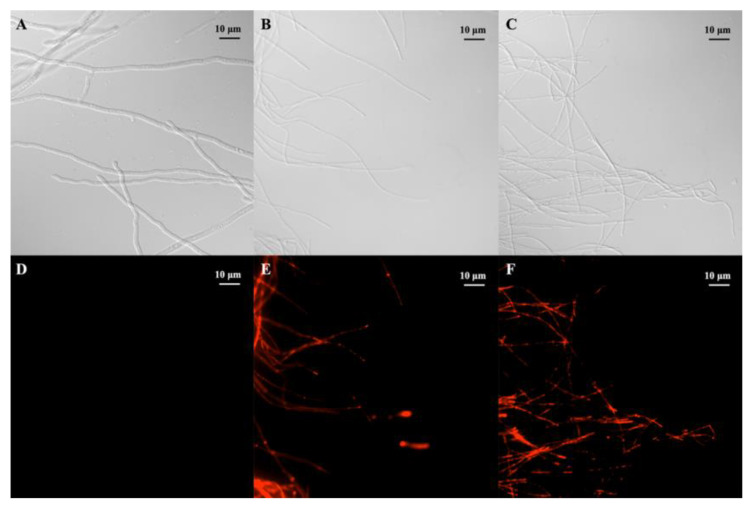
Morphology of G. saubinetii treated with **E_13_** at different dosages under optical microscopy and fluorescence microscopy. (**A**–**C**: bright field; **D**–**F**: fluorescence field. **A**,**D** 0 mg/L (CK); **B**,**E**: 2 eq. EC_50_; **C**,**F**: 4 eq. EC_50_).

**Figure 7 ijms-24-08898-f007:**
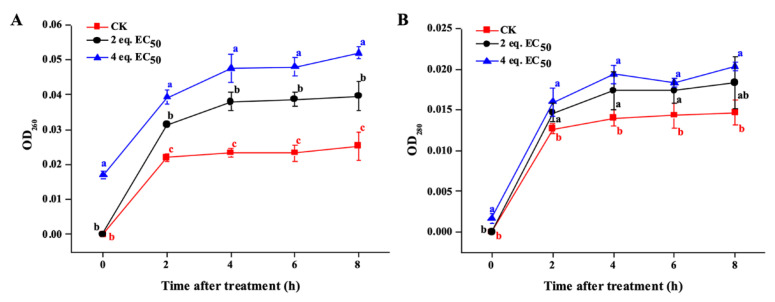
Leakage of cytoplasmic contents of *G. saubinetii* mycelia with **E_13_** treatment. (**A**) nucleic acid concentration; (**B**) protein concentration. ^a,b,c^ Different lowercase letters in a column indicate significant differences between mean values evaluated by Duncan’s multiple range test (*p* < 0.05).

**Table 1 ijms-24-08898-t001:** Inhibition effect of title compounds against seven pathogenic fungi at 100 mg/L ^a^.

Compound No.	Inhibition Rate ± SD (%)						
	GS	VD	SS	FO	FP	TC	PC
**E_1_**	69.9 ± 2.0	73.8 ± 3.0	88.8 ± 2.3	59.1 ± 0.9	57.1 ± 0.5	65.8 ± 0.8	29.2 ± 1.3
**E_2_**	69.6 ± 1.1	69.4 ± 0.9	71.8 ± 3.9	10.9 ± 2.7	30.3 ± 0.8	45.8 ± 1.7	39.8 ± 0.5
**E_3_**	0	16.1 ± 0.5	11.3 ± 0.5	0	11.2 ± 0.5	48.3 ± 2.2	11.1 ± 1.8
**E_4_**	27.5 ± 2.0	22.1 ± 0.5	21.7 ± 4.0	0	15.6 ± 0.8	25.3 ± 2.4	23.7 ± 0.9
**E_5_**	24.5 ± 0.0	22.4 ± 1.1	11.9 ± 1.9	54.2 ± 0.5	58.2 ± 0.8	84.9 ± 0.5	36.6 ± 3.3
**E_6_**	64.1 ± 0.6	93.8 ± 1.9	49.4 ± 0.5	9.4 ± 4.1	33.1 ± 0.5	67.5 ± 1.7	32.8 ± 2.2
**E_7_**	75.3 ± 2.9	80.3 ± 2.8	72.4 ± 1.7	35.8 ± 0.5	56.6 ± 0.8	81.4 ± 1.3	62.3 ±0.9
**E_8_**	64.1 ± 1.0	50.6 ± 0.5	83.1 ± 2.6	11.2 ± 0.5	28.1 ± 1.0	24.7 ± 1.7	17.3 ± 0.5
**E_9_**	59.8 ± 0.5	83.0 ± 1.9	92.8 ± 0.5	8.2 ± 0.9	38.9 ± 1.3	19.7 ± 2.6	26.9 ± 1.3
**E_10_**	78.5 ± 0.3	91.8 ± 2.2	87.5 ± 0.0	69.4 ± 1.4	68.3 ± 0.5	81.1 ± 1.1	53.8 ± 1.0
**E_11_**	65.7 ± 0.0	75.2 ± 0.5	68.8 ± 2.7	8.8 ± 2.3	21.0 ± 3.1	41.4 ± 5.4	5.0 ± 0.5
**E_12_**	48.7 ± 0.6	65.2 ± 0.5	54.5 ± 1.8	33.3 ± 1.4	38.3 ± 0.5	28.1 ± 1.3	46.5 ± 0.9
**E_13_**	78.3 ± 0.9	100	92.5 ± 0.8	70.3 ± 0.5	69.7 ± 0.8	87.5 ± 0.0	76.3 ± 1.5
**E_14_**	80.2 ± 0.9	95.9 ± 1.4	71.9 ± 3.8	68.8 ± 1.1	69.1 ± 1.3	66.9 ± 0.5	70.2 ± 1.8
**E_15_**	37.9 ± 0.6	23.6 ± 0.0	34.2 ± 1.4	27.3 ± 1.6	49.5 ± 2.4	60.8 ± 1.7	61.1 ± 2.0
**E_16_**	42.3 ± 1.3	39.3 ± 1.0	24.9 ± 1.1	27.0 ± 1.0	31.8 ± 0.8	49.2 ± 0.7	35.1 ± 0.9
**E_17_**	69.5 ± 2.2	83.3 ± 2.2	89.7 ± 0.5	10.0 ± 1.8	12.3 ± 0.7	18.1 ± 0.5	58.2 ± 1.0
**E_18_**	73.3 ± 1.7	81.3 ± 1.8	69.0 ± 0.9	0	24.3 ± 2.5	48.1 ± 0.5	37.7 ± 1.5
**E_19_**	66.0 ± 2.5	67.9 ± 0.0	69.7 ± 0.5	52.4 ± 1.0	39.3 ± 0.9	57.3 ± 0.8	48.8 ± 0.5
**E_20_**	76.0 ± 1.9	94.4 ± 0.0	80.5 ± 0.5	69.0 ± 2.5	69.7 ± 1.7	65.9 ± 1.0	74.2 ± 0.9
**E_21_**	65.1 ± 0.6	81.8 ± 1.9	77.3 ± 0.5	24.9 ± 2.6	60.9 ± 0.5	61.2 ± 1.1	8.5 ± 1.0
**E_22_**	71.6 ± 1.1	64.2 ± 0.6	67.9 ± 0.5	--	--	--	--
**E_23_**	72.2 ± 1.0	80.6 ± 0.6	52.5 ± 0.5	--	--	--	--
**E_24_**	74.1 ± 1.0	81.6 ± 0.6	65.1 ± 2.8	--	--	--	--
**E_25_**	71.9 ± 0.6	78.2 ± 1.6	57.1 ± 1.1	--	--	--	--
**E_26_**	73.8 ± 0.6	80.9 ± 1.0	61.7 ± 1.4	--	--	--	--
**E_27_**	74.4 ± 2.0	68.6 ± 1.5	74.1 ± 0.9	--	--	--	--
**E_28_**	76.0 ± 1.0	74.6 ± 0.6	67.9 ± 0.5	--	--	--	--
**MP**	19.1 ± 0.4	27.2 ± 0.9	14.0 ± 0.5	26.5 ± 0.9	27.5 ± 1.1	23.6 ± 1.9	25.2 ± 1.0
**HY**	--	--	--	--	--	79.3 ± 1.6	69.6 ± 0.5

^a^ Values are means ± SD of three replicates. “--” not test. GS: Gibberella saubinetii (Durieu and Mont.) Sacc.; VD: Verticillium dahliae Kleb.; SS: Sclerotinia sclerotiorum (Lib.) de Bary; FO: Fusarium oxysporum f. sp. Cucumerinum; FP: Fusarium proliferatum (Matsush.) Nirenberg ex Gerlach and Nirenberg; TC: Thanatephorus cucumeris (A.B. Frank) Donk; PC: Phytophthora capsici Leonian; MP: mandipropamid; HY: hymexazol. MP and HY were pure compounds.

**Table 2 ijms-24-08898-t002:** EC_50_ values of some title compounds against five kinds of pathogenic fungi ^a^.

Compound No.	EC_50_ (mg/L)				
	GS	VD	SS	TC	PC
**E_1_**	47.4 ± 2.6	50.7 ± 2.8	23.0 ± 1.5	--	--
**E_2_**	37.3 ± 1.4	37.2 ± 1.2	27.1 ± 1.5	--	--
**E_5_**	--	--	--	18.5 ± 0.4	--
**E_6_**	49.9 ± 0.4	12.7 ± 0.5	90.5 ± 3.5	--	--
**E_7_**	30.6 ± 3.0	14.3 ± 0.1	40.1 ± 2.7	5.7 ± 0.2	--
**E_8_**	79.1 ± 2.6	>100	13.8 ± 3.1	--	--
**E_9_**	24.6 ± 0.7	29.0 ± 2.1	10.3 ± 1.0	--	--
**E_10_**	29.4 ± 0.6	37.2 ± 1.0	21.9 ± 0.4	36.3 ± 1.3	--
**E_13_**	20.4 ± 1.3	18.5 ± 1.2	33.5 ± 1.8	7.1 ± 0.1	42.9 ± 0.7
**E_14_**	21.5 ± 1.0	23.1 ± 0.8	37.6 ± 1.9	--	49.3 ± 1.4
**E_17_**	22.0 ± 1.5	16.1 ± 0.2	27.9 ± 0.2	--	--
**E_18_**	24.5 ± 0.9	15.8 ± 1.4	8.0 ± 0.3	--	--
**E_19_**	56.1 ± 0.9	65.7 ± 0.7	47.8 ± 1.7	--	--
**E_20_**	31.6 ± 0.6	29.6 ± 2.3	39.1 ± 1.1	--	50.9 ± 1.4
**E_21_**	27.3 ± 2.0	13.4 ± 0.6	36.3 ± 0.3	37.6 ± 1.0	--
**E_22_**	51.1 ± 0.2	61.3 ± 1.6	48.5 ± 0.4	--	--
**E_23_**	32.5 ± 0.9	32.3 ± 0.4	81.4 ± 2.1	--	--
**E_24_**	30.8 ± 1.5	38.5 ± 1.0	51.6 ± 5.4	--	--
**E_25_**	31.3 ± 1.5	33.7 ± 3.5	75.2 ± 3.4	--	--
**E_26_**	32.7 ± 1.1	30.4 ± 1.3	54.0 ± 1.8	--	--
**E_27_**	25.2 ± 1.0	43.0 ± 1.6	37.2 ± 0.8	--	--
**E_28_**	24.2 ± 0.4	26.9 ± 0.2	47.9 ± 0.2	--	--
**MP**	>100	>100	>100	--	--
**HY**	--	--	--	13.8 ± 1.5	17.9 ± 1.3

^a^ Values are means ± SD of three replicates. “--” not test.

## Data Availability

All data generated in this study is presented in the current manuscript. No new datasets were generated. Data is available upon request from the corresponding author.

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
