# Peer review of "Design, Synthesis, In Vitro Antifungal Activity and Mechanism Study of the Novel 4-Substituted Mandelic Acid Derivatives"

_ijms, 2023, doi:10.3390/ijms24108898_

Round 1

Reviewer 1 Report

The evaluated work contributes a lot of information regarding the use of mandelic acid derivatives as a potential fungicide. The authors' results indicate the possible use of the synthesised substance as a component of a commercial preparation. I have a few comments for the authors:

1/ The order of the chapters in the paper should be respected. First the chapter describing the material and methods adopted and only then the chapter containing the results.

2/ Mandipropamide was used as a control subject. There is no information on whether this was a chemical compound in pure form or as a component of a fungicide. This should be clarified

3/ Species of fungi pathogenic to crop plants were tested. I propose to keep their current nomenclature according to the systematics adopted after Index Fungorum (https://www.indexfungorum.org/names/names.asp).

Current names and full descriptions of pathogenic species according to Index Fugorum are:

Gibberella saubinetii - current name: Fusarium graminearum Schwabe,

Verticillium dahlia – should be: Verticillium dahliae Kleb.

Sclerotinia sclerotiorum - Sclerotinia sclerotiorum (Lib.) de Bary,,

Fusarium oxysporum - Fusarium oxysporum Schltdl.

Fusarium prolifeatum – should be: Fusarium proliferatum (Matsush.) Nirenberg ex Gerlach & Nirenberg,

Thanatephorus cucumeris - current name Thanatephorus cucumeris (A.B. Frank) Donk

The chemical compound synthesis routes are only shown in the figure - there is no detailed description of the syntheses and the efficiency of the individual stages. In this form, the syntheses are unrepeatable. This should be supplemented.

Author Response

Detailed Responses to Reviewers’ comments:

Review 1:

The evaluated work contributes a lot of information regarding the use of mandelic acid derivatives as a potential fungicide. The authors' results indicate the possible use of the synthesised substance as a component of a commercial preparation. I have a few comments for the authors:

1.The order of the chapters in the paper should be respected. First the chapter describing the material and methods adopted and only then the chapter containing the results.

Answer: Thanks for your suggestions. The order of the chapters in this manuscript has been arranged according to the IJMS’s requirements. According to IJMS's rules, the correct sequence of sections includes the following: 1. Introductions; 2. Results; 3. Discussion (2 and 3 can be combined as : 2. Results and Discussion); 4. Materials and Methods; 5. Conclusions (Optional).

2.Mandipropamide was used as a control subject. There is no information on whether this was a chemical compound in pure form or as a component of a fungicide. This should be clarified.

Answer: Thanks for your kind suggestions. The control fungicides mandipropamid and hymexazol used in this manuscript are all pure compounds, as is explained in the text.

3.Species of fungi pathogenic to crop plants were tested. I propose to keep their current nomenclature according to the systematics adopted after Index Fungorum (https://www.indexfungorum.org/names/names.asp).

Current names and full descriptions of pathogenic species according to Index Fugorum are:

Gibberella saubinetii - current name: Fusarium graminearum Schwabe,

Verticillium dahlia – should be: Verticillium dahliae Kleb.

Sclerotinia sclerotiorum - Sclerotinia sclerotiorum (Lib.) de Bary,

Fusarium oxysporum - Fusarium oxysporum Schltdl.

Fusarium prolifeatum – should be: Fusarium proliferatum (Matsush.) Nirenberg ex Gerlach & Nirenberg,

Thanatephorus cucumeris - current name Thanatephorus cucumeris (A.B. Frank) Donk

Answer: Thanks for your kind suggestions. We haved checked and revised all the nomenclatures in this manuscript according to the systematics adopted after Index Fungorum (https://www.indexfungorum.org/names/names.asp). The “Verticillium dahlia” has been revised into “Verticillium dahliae Kleb.”. The “Sclerotinia sclerotiorum” has been revised into “Sclerotinia sclerotiorum (Lib.) de Bary”. The “Fusarium oxysporum” has been revised into “Fusarium oxysporum f. sp. Cucumerinum.”. The “Fusarium prolifeatum”  has been revised into “Fusarium proliferatum (Matsush.) Nirenberg ex Gerlach & Nirenberg”. The “Thanatephorus cucumeris” has been revised into “Thanatephorus cucumeris (A.B. Frank) Donk”. The “Phytophthora capsici” has been revised into “Phytophthora capsici Leonian”. The “Gibberella saubinetii” has been revised into “Gibberella saubinetii (Durieu & Mont.) Sacc. (Fusarium graminearum Schwabe)”.

4.The chemical compound synthesis routes are only shown in the figure - there is no detailed description of the syntheses and the efficiency of the individual stages. In this form, the syntheses are unrepeatable. This should be supplemented.

Answer: Thanks for your kind suggestions. The chemical compound synthesis route in Figure 3 has been revised. The detailed steps for the synthesis of intermediates B, C, D, and target compounds are described in the section of “Materials and Methods”. The yields of the target compounds are provided in the Supplementary Materials.

Reviewer 2 Report

Chen et al. developed Novel 4-Substituted Mandelic Acid Derivatives as effective antifungal agents. The authors conducted antifungal activity against Gibberella saubinetii, Verticillium dahlia, Sclerotinia sclerotiorum, and Thanatephorus cucumeris. They also investigated the mechanism of action of these antifungal agents.

I have some minor concerns that need to be addressed before publication.

1.      Antifungal activity and mechanism of action of hit molecule should be discussed in detail.

2.      Structure-activity relationship (SAR) should be discussed in detail.

3.      HPLC purity of target compounds should be provided.

4.      The toxicity study of these compounds should be addressed.

5.      Typo and grammatical errors should be addressed. 

Typo and grammatical errors should be addressed

Author Response

Chen et al. developed Novel 4-Substituted Mandelic Acid Derivatives as effective antifungal agents. The authors conducted antifungal activity against Gibberella saubinetii, Verticillium dahlia, Sclerotinia sclerotiorum, and Thanatephorus cucumeris. They also investigated the mechanism of action of these antifungal agents.

I have some minor concerns that need to be addressed before publication.

1.Antifungal activity and mechanism of action of hit molecule should be discussed in detail.

Answer: Thanks for your kind suggestions. The antifungal activity and mechanism of action of target compounds have discussed in detail. To obtain high-antifungal-activity compounds with novel action mechanisms, a series of 4-substituted mandelic acid derivatives containing a 1,3,4-oxadiazole moiety were designed and synthesized. In vitro bioassay results revealed that some compounds exhibited excellent activity against the tested fungi. As indicated, some title compounds exhibited excellent inhibitory activities against the seven tested fungi at a concentration of 100 mg/L. In detail, compounds E10, E13 and E14 showed good antifungal activity against G. saubinetii, with inhibition rates of 78.5%, 78.3% and 80.2%, respectively, which were higher than mandipropamid (19.1%). The inhibition rates of E6, E10, E13, E14 and E20 against V. dahliae were 93.8%, 91.8%, 100%, 95.9% and 94.4%, respectively, which were much higher than mandipropamid (27.2%). Compounds E9, E13 and E17 exhibited 92.8%, 92.5% and 89.7% inhibition of S. sclerotiorum, respectively, which was much better than that of the commercialized fungicide mandipropamid (14.0%). Compounds E10, E13 and E14 showed good antifungal activity against Fusarium oxysporum, with inhibition rates of 69.4%, 70.3% and 68.8%, respectively, which were higher than mandipropamid (26.5%). Compounds E5, E7, E10 and E13 showed remarkable antifungal activity against T. cucumeris, with inhibition rates of 84.9%, 81.4%, 81.1% and 87.5%, respectively, which were higher than hymexazol (79.3%). Among them, the EC50 values of E13 against Gibberella saubinetii, E6 against Verticillium dahlia and E18 against Sclerotinia sclerotiorum were 20.4, 12.7, and 8.0 mg/L, respectively, which were highly superior to that of the commercialized fungicide mandipropamid. In order to investigate the antifungal mechanism of the highly active compounds, we conducted morphological studies using compound E13. A morphological study of G. saubinetii with fluorescence microscope (FM) and scanning electron microscope (SEM) indicated that E13 broke the surface of the hyphae and destroyed cell membrane integrity with increased concentration, thereby inhibiting fungal reproduction. Further cytoplasmic content leakage determination results showed a dramatic increase of the nucleic acid and protein concentrations in mycelia with E13 treatment, which also indicated that the title compound E13 can destroy cell membrane integrity and affect the growth of fungi. These results provide important information for further study of the mechanism of action of mandelic acid derivatives and their structural derivatization.

  1. Structure-activity relationship (SAR) should be discussed in detail.

Answer: Thanks for your kind suggestions. We discussed the Structure-activity relationship (SAR) in detail. The preliminary structure–activity relationship analysis showed that when R2 were the same substituted phenyl groups and R1 were electron withdrawing groups, the anti-G. saubinetii and anti-S. sclerotiorum activities were higher than electron-donating groups, such as E9 > E2, E13/E20 > E6, E14/E21 > E7. When R1 was the same and R2 is substituted phenyl group, the anti-G. saubinetii and anti-S. sclerotiorum activities were higher than that of non-substituted phenyl groups, such as E2/E7 > E1, E9/E10/E13/E14 > E8. In addition, when R1 is a halogen group and the para-substituent group or ortho-substituted group of R2 are halogen groups, the anti-G. saubinetii and anti-S. sclerotiorum activities were higher than electron-donating groups, such as E13/E14 > E9, E24/E25 > E22.

  1. HPLC purity of target compounds should be provided.

Answer: Thanks for your suggestions. On the basis of 1H NMR, 13C NMR and HRMS to identify the structures of all compounds, the purity of these compounds was tested by HPLC, and the results were provided in the Supplementary Materials.

  1. The toxicity study of these compounds should be addressed.

Answer: Thanks for your kind suggestions. In this manuscript, we synthesized a series of 4-substituted mandelic acid derivatives containing a 1,3,4-oxadiazole moiety and found some compounds with much higher antifungal activity than the control agents. For the further derivatization of this skeleton, it is meaningful to conduct toxicological studies on these highly active compounds, which is exactly what we will do next.

  1. Typo and grammatical errors should be addressed.

Answer: Thank you very much for your kind comments. The whole manuscript has been carefully checked and revised with respect to typo and grammatical errors, which are all marked in red in the revised manuscript.
